# Structural Analysis of Cardanol and Its Biological Activities on Human Keratinocyte Cells

**DOI:** 10.3390/metabo15020083

**Published:** 2025-01-30

**Authors:** Shereen Basiouni, Nina Abel, Wolfgang Eisenreich, Helen L. May-Simera, Awad A. Shehata

**Affiliations:** 1Institute of Molecular Physiology, Johannes Gutenberg University, 55128 Mainz, Germany; sbasioun@uni-mainz.de (S.B.); may-simera@uni-mainz.de (H.L.M.-S.); 2Structural Membrane Biochemistry, Bavarian NMR Center, Technical University of Munich (TUM), 85748 Garching, Germany; ge89beg@mytum.de (N.A.); awad.shehata@tum.de (A.A.S.)

**Keywords:** cashew, cardanol, CNSL, cytotoxicity, genotoxicity, wound healing, fatty acids

## Abstract

**Background/Objectives:** Cashew nutshell liquid (CNSL) is obtained during the industrial processing of cashew nuts. It contains anacardic acid (2-hydroxy-6-n-pentadecylbenzoic acid) and cardanol (3-n-pentadecylphenol). Therefore, CNSL provides a rich source of phenolic lipids serving as natural antioxidants or precursors for industrial uses. Here, we have analyzed in detail a commercial sample of cardanol by nuclear magnetic resonance (NMR) spectroscopy and its biological activities in the human keratinocyte cell line (HaCaT cells). **Methods:** The cytotoxic effects, genotoxicity, cell proliferation, and healing properties on HaCaT cells were studied using the 3-(4,5-dimethyl-2-thiazolyl)-2,5-diphenyl-2-H-tetrazolium bromide (MTT) assay, comet assay, proliferation assay, and scratch assay, respectively. Additionally, the modulatory effect of cardanol on the cellular fatty acid profile of HaCaT cells was analyzed by gas chromatography. **Results:** NMR showed the structure of cardanol as a mixture of the 8′-monoene (42%), the 8′,11′-diene (22%), and the 8′,11′,14′-triene (36%) for the pentadecyl side chain with all double bonds in Z configuration. The cytotoxic effects on HaCaT cells only occurred at high concentrations of cardanol (>10 µg/mL), which caused significant reductions in cell viability. Using the comet assay, a dose-dependent increase in DNA damage was found at concentrations above 10 µg/mL. Scratch assays revealed that cardanol achieved 99% wound closure of HaCaT cells treated with 1 µg/mL cardanol after 48 h. Cardanol at 1 and 0.1 µg/mL significantly enhanced HaCaT cell proliferation and promoted migration, contributing to accelerated wound healing processes. As shown by gas chromatography, 1 µg/mL cardanol increased the total amount of polyunsaturated fatty acids (PUFA), including ω-3, ω-6, and ω-9 fatty acids. **Conclusions:** Together, these findings suggest that concentrations of <10 µg/mL cardanol are safe and exhibit beneficial biological activities, particularly wound-healing effects on HaCaT cells. Further studies are necessary to explore additional potential applications of cardanol, to refine its formulations for clinical use, and to ensure its safety and action in other target cells and species.

## 1. Introduction

Natural product-derived molecules have been extensively studied for their various applications in health and disease treatment due to their versatility, safety, and cost-effectiveness [1,2,3,4]. These substances can also be beneficial in treating various skin conditions by promoting wound healing [3] and improving skin barrier repair thanks to their anti-inflammatory and antioxidant properties [4]. Among these natural bioactive substances are products from cashew *Anacardium occidentale* L., a tropical tree that belongs to the *Anacardiaceae* family.

The cashew tree originated in northeastern Brazil and is now grown in several countries, including India, Mozambique, Tanzania, Kenya, Vietnam, Indonesia, and Thailand. It is primarily cultivated for its nuts. The production processes of cashew nuts generate substantial amounts of cashew nutshell liquid (CNSL), which is produced on a large scale using various methods, such as cold press extraction and solvent extraction [5,6], thermomechanical processes, and supercritical CO_2_ extraction [7]. The bioactive molecules in CNSL are influenced by the extraction method used. In the industry, three methods are currently used for extracting CNSL from cashew nut shells: thermal extraction, mechanical pressing, and solvent extraction [5,6]. Based on the extraction method, CNSL can be classified into natural and technical CNSL [8].

Natural CNSL is obtained through solvent extraction and mechanical pressing of the shells [9]. This form of CNSL is valued for its diverse biological activities, primarily attributed to anacardic acid (2-hydroxy-6-n-pentadecylbenzoic acid), which possesses antimicrobial, fungicidal, insecticidal, termiticidal, antioxidant, and enzymatic inhibition properties. In contrast, technical CNSL is produced by heating natural CNSL to temperatures above 180–200 °C [8]. This heating process decarboxylates the thermolabile anacardic acid, converting it into cardanol (3-n-pentadecylphenol) and increasing its concentration to 60–65%. The higher cardanol content makes technical CNSL a valuable raw material for various industrial applications, including friction linings, paints, varnishes, laminates, epoxy resins, foundry chemicals, plastic formulations, and as an antioxidant in biodiesel [10].

CNSL, rich in anacardic acid, cardanol, and cardol (5-n-pentadecylresorcinol), has also garnered significant interest in medicinal chemistry because it combines the structural motives of phenolics and unsaturated fatty acids that could interact with biological targets through various pathways. These pathways include anti-inflammatory and antioxidant [11,12,13], antibacterial [14], and anti-carcinogenic [15] effects, as well as larvicidal and anticholinesterase activities [13].

Recently, CNSL has also been studied as a natural feed additive in animal production. In broilers, CNSL demonstrated performance and slaughter yields comparable to the growth promoter virginiamycin while reducing the concentration of *Escherichia coli* in the intestinal contents [16]. Furthermore, CNSL has been shown to reduce methanogenesis in vitro [17] and in vivo [18]. It may also modulate the microbiota in chickens, leading to beneficial shifts in fermentation profiles, including the production of short-chain fatty acids and ammonia, which may improve hindgut health, enhancing animal health, productivity, and overall performance. Hosokawa and colleagues reported that CNSL exhibits an inhibitory effect against *Clostridium perfringens* [19], further supporting its antimicrobial potential. Moreover, the bioactive compounds in CNSL exhibit anti-inflammatory and antimicrobial properties, positioning it as a promising alternative to conventional growth promoters in livestock nutrition [20].

Despite the numerous documented beneficial effects of CNSL, it can also have irritating effects. Andonaba and others reported skin damage and esthetic issues among workers handling cashew nuts at shelling stations [21]. To date, there is a gap in research specifically addressing cardanol’s safety profile and its therapeutic potential for human skin cells, particularly in wound healing. In this study, we have therefore focused on cardanol and its unexplored properties in skin treatment. For this purpose, we have used the human keratinocyte cell line (HaCaT) to assess the safety of cardanol by investigating its cytotoxic and genotoxic effects, alongside its potential wound-healing effects and its impact on cellular lipid profiles.

## 2. Materials and Methods

### 2.1. Cells and Materials

The human keratinocyte cell line (HaCaT cells), obtained from CLS Cell Lines Service (Eppelheim, Baden-Württemberg, Germany), was cultured in Dulbecco’s Modified Eagle Medium (DMEM) (Gibco Life Technologies, Darmstadt, Germany). This medium was supplemented with 10% fetal bovine serum (FBS) (Cytiva, Freiburg, Germany), along with 1% (*v*/*v*) penicillin/streptomycin and 2 mM L-glutamine. The cells were sub-cultured at 37 °C in an atmosphere containing 5% CO_2_. Cardanol was obtained from Sona Meta Chem Pvt. Ltd., Pondicherry, India. According to the manufacturer, cardanol was obtained by the vacuum distillation of CNSL. Cardanol 50% was prepared in castor oil (Ethoxylated Castor Oil, Castor International, Deurne, the Netherlands) to analyze its biological activities on cell cultures. DMEM supplemented with castor oil was used as a basic control medium.

### 2.2. NMR Spectroscopy Analysis of Cardanol

Cardanol was analyzed by NMR (Bruker AVANCE 500 spectrometers, Rheinstetten, Germany). Approximately 30 mg of the brownish-colored liquid was dissolved in 0.5 mL of CDCl_3_. One- and two-dimensional ^1^H (500 MHz) and ^13^C (126 MHz) NMR spectra were recorded at 27 °C. ^1^H-Detected experiments (double-quantum-filtered COSY, NOESY with a mixing time of 1 s, DEPT-edited HSQC, and HMBC) were performed with an inverse ^1^H/^13^C probe head, while direct ^13^C measurements (^1^H-decoupled ^13^C-spectra, DEPT135, and DEPT90) were conducted using a QNP ^13^C/31P/29Si/19F/1H cryoprobe. All experiments used standard parameter sets from the TOPSPIN 3.2 software package (Bruker, Germany). Data were processed using MNova software (Version 14.1.0.-24037; Mestre Lab Research, Santiago de Compostela, Spain). Signal assignments were based on the observed signals in the proton–proton and proton–carbon correlation experiments.

### 2.3. Cytotoxicity Assay

The 3-(4,5-dimethyl-2-thiazolyl)-2,5-diphenyl-2-H-tetrazolium bromide (MTT) assay was used to evaluate the cytotoxicity of cardanol on HaCaT cells. Briefly, HaCaT cells were cultivated in 96-well plates at a seeding density of 5.000 cells per well, allowing for incubation until cells reached approximately 70–80% confluency. Subsequently, the medium was aspirated, and fresh medium containing varying concentrations of cardanol (100, 10, 1, 0.1, 0.01 µg/mL) was added. After 24 h of incubation, an MTT assay was conducted following the manufacturer’s guidelines. In brief, 10 µL of a 5 mg/mL MTT solution (Roche, Mannheim, Germany) was introduced to each well, and the cultures were incubated at 37 °C for 4 h. Post-incubation, 100 µL of MTT buffer (Roche, Mannheim, Germany) was employed to dissolve the Formazan and incubated overnight at 37 °C. The optical density, measured spectrophotometrically at 570 nm using a microplate reader (Tecan, Tecan Trading AG, Männedorf, Switzerland), allowed for normalizing cell viability percentages relative to the untreated control.

### 2.4. Genotoxicity Assay

Comet assays were performed to assess the genotoxicity on HaCaT cells using single-cell gel electrophoresis (R&D Systems, Minneapolis, MN, USA). Briefly, HaCaT cells were seeded at a density of 0.3 × 10^6^ cells per well in a 6-well plate in medium supplemented with cardanol (100, 10, 1, 0.1, 0.01 µg/mL) and cultured for 24 h. Basic media supplemented with 10 µg/mL of castor oil were used as a negative control. Cells were washed with PBS, trypsinized, collected by centrifugation at 300× *g* for 5 min at 4 °C, and then suspended in ice-cold PBS. Cell counting was performed using the Trypan blue assay, and the cells were diluted to a concentration of 1 × 10^5^ cells/mL. The cell suspension was mixed with low-melting agarose (37 °C) at 1:10 (*v*/*v*) and immediately applied onto prewarmed 37 °C comet slides. After incubating in the dark at 4 °C for 30 min, the slides underwent sequential treatment with lysis solution and neutral electrophoresis buffer for 60 min and 30 min, respectively, at 4 °C. Subsequently, single-cell electrophoresis was carried out for 45 min at 4 °C with a potential difference of 1 V/cm. The slides were then subjected to DNA precipitation solution treatment for 30 min at room temperature, followed by dehydration in 70% ethanol for 30 min at room temperature and drying at 37 °C for 15 min. The dried slides were placed in a desiccator overnight to bring the cells into a single plane for easy observation. Staining was performed using 0.003% SYBR Gold staining solution for 30 min at room temperature in the dark. After removing the excess SYBR solution, the slides were dipped in water and thoroughly dried at 37 °C before scoring. Image capture was performed using a Leica DM6000 B inverted fluorescence microscope with the FITC 488 filter (Leica, Wetzlar, Germany). Image processing utilized Fiji software (National Institutes of Health, Bethesda, MD, USA) with the OpenComet v1.3 plug-in. Quantification focused only on individual cells rather than clustered cells. As the positive control, HaCaT cells treated with a known DNA-damaging agent, hydrogen peroxide (100 μM; 20 min at 4 °C), were included. The olive tail moment was employed as a metric to evaluate DNA damage [22].

### 2.5. Wound-Healing Assay

To estimate the wound healing in HaCaT cells, the scratch assay was performed according to [23]. Briefly, HaCaT cells were seeded in 12-well plates at 0.5 × 10^6^ cells/mL. After reaching confluency, the cell monolayer was scraped with a 200 μL pipette tip across the well. Cells were washed with PBS and replaced with fresh medium containing different concentrations (10, 1, and 0.1 µg/mL) of cardanol. The scratched region was photographed immediately and every 24 h after scratching using a Leica DMI300 B microscope (Leica microsystems) at 10× magnification. The total scratch area was analyzed using ImageJ software (National Institutes of Health, Bethesda, MD, USA), and the wound closure was calculated based on the following equation.$$Wound\;closure\%=\frac{Wd0-Wdt}{Wd0}\times 100$$
Wd0 is the distance between wound boundaries immediately after the wounding procedure, and Wdt is the distance between wound boundaries after time “t” of sample treatment.

### 2.6. Proliferation Assay

The CyQUANT^®^ cell proliferation reagent assay (Thermofischer, Waltham, MA, USA) was utilized as a high-throughput method for assessing cell proliferation. HaCaT cells, plated at a density of 5 × 10^3^ cells per well in a 96-well plate, underwent treatment with cardanol at concentrations of 10, 1, and 0.1 µg/mL for 24 h. Following incubation, cells were washed twice with PBS (pH 7.4). The subsequent labeling of cells with the CyQUANT^®^ reagent followed the manufacturer’s instructions. Fluorescence readings were obtained using a Tecan microplate reader (Tecan Trading AG, Männedorf, Switzerland), with excitation and emission wavelengths set at 480 nm and 520 nm, respectively. The results were expressed as a percentage of the control value.

### 2.7. Fatty Acid Profiles of HaCaT Cells

HaCaT cells were supplemented with 1 µg/mL cardanol or the basic medium (containing 1 µg castor oil) and incubated for 24 h. The analysis of fatty acid composition involved a sequential process. Initially, lipid extraction was conducted, and gas chromatography was performed, as outlined in [24]. To trans-esterify the membrane lipids, a mixture comprising 500 µL of methanolic HCl, 250 µL of n-hexane, and 500 µL of an internal standard (0.8 mg di-C17-phosphatidylcholine dissolved in 1 mL methanol with 0.2% butylhydroxytoluol as an antioxidant) was employed. After cooling, 500 µL of n-hexane and 1 mL of distilled water were added. The upper hexane phase, containing the trans-esterified products, was collected and subjected to evaporation using nitrogen gas. The resulting fatty acid methylesters (FAMEs) were reconstituted in 60 µL of n-hexane. Following this, a 1 μL aliquot of the FAME solution was injected on-column into a Varian CP 3800 gas chromatograph (Varian, Darmstadt, Germany) equipped with an Omegawax TM 320 column (0.32 mm internal diameter, 30 m length) (Supelco, Bellefonte, PA, USA). The column temperature was maintained at 200 °C throughout the analysis.

### 2.8. Statistical Analysis

Data are shown as means with standard deviation (SD). A one-way analysis of variance followed by Student’s unpaired *t*-test was used to identify significant differences between the means. The statistical analysis was conducted using GraphPad Prism 4 (La Jolla, CA, USA). In all instances, *p* < 0.05 was assumed to indicate significant differences.

## 3. Results

### 3.1. NMR Analysis

The ^1^H-NMR and ^13^C-NMR spectra of the commercial cardanol sample showed well-resolved signals (for the ^1^H-NMR spectrum, see Figure 1). Most NMR signals could be assigned based on the observed correlations in double-quantum-filtered COSY, NOESY, DEPT-edited HSQC, and HMBC-spectra (Figure 2, Appendix A). Although the ^1^H-NMR signals for H-2 and H-6 partially overlapped, the detected chemical shifts, integrals, couplings, and correlations in the COSY, NOESY, HSQC, and HMBC experiments (Figure 2, Appendix A) were fully in line with 3-alkylphenols with the ^1^H-NMR signals of the phenol ring at 6.71 ppm for H-2 (s), 6.81 ppm for H-4 (d), 7.18 ppm for H-5 (t), and 6.71 ppm for H-6 (d) (for atom numbering, see Figure 1 and Appendix A). The ^1^H spin systems were determined from the COSY experiment (Figure 2). The ^13^C signals could be assigned by DEPT-edited HSQC and HMBC experiments (Figure 2). The connectivity between C-3 of the phenol ring with the alkyl side chains was especially supported by HMBC correlations between the signals of ^13^C-3 and H-1′/H-2′. Notably, signals for 2,3-disubstituted phenol rings, such as 2-carboxy-3-alkylphenol in anacardic acid or 5-hydroxy-3-alkylphenol in cardol, or for other phenolics, could not be detected. The full analysis of the side chains was more complex and reflected a mixture of pentadecyl moieties as a monoene, diene, and triene (C_15_H_25-31_, numbered C1′ to C15′), respectively. We have designated these three constituents of cardanol as components A, B, and C, respectively (Figure 1 and Appendix A).

### 3.2. Cytotoxicity of Cardanol on HaCaT Keratinocytes

Cardanol was found to be safe for HaCaT cells based on the MTT assay. The cytotoxic effect was observed only at high doses (100 µg/mL). Treatment with ≤10 µg/mL cardanol did not affect HaCaT viability. However, cell death was significant at 100 µg/mL (*p* < 0.0001) (Figure 3). As a control, castor oil did not show cytotoxic effects at all the tested doses (100, 10, 1, 0.1, and 0.01 µg/mL).

### 3.3. Genotoxicity (Comet Assay)

The toxicity of the cardanol was evaluated by the comet assay performed in the alkaline variant (Figure 4). The mean olive tail moment measurements of HaCaT cells subjected to H_2_O_2_ exposure as a positive control for 30 min, both in the absence and presence of 25 µM H_2_O_2_ (*p* < 0.0001). A notable rise in tail moment values was evident following treating cells with cardanol at 10 µg/mL (Figure 4). However, no significant effects have been found in HaCaT cells supplemented with low doses (1 and 0.1 µg/mL). Automated tail moment calculations revealed that the supplementation of HaCaT cells with cardanol at concentrations of 1 and 0.1 µg/mL did not significantly impact the DNA fragmentation compared to the untreated control (Figure 4).

### 3.4. Wound Healing Properties of Cardanol

Using the scratch assay, the effects of cardanol on enhancing the migration of HaCaT cells are shown in Figure 5 and Figure 6. At a concentration of 1 µg/mL, cardanol induced a significant effect (*p* < 0.05) after 24 h of incubation. The wound area showed a significant reduction, smaller than the initial wound gap (*p* < 0.01) after 48 h of treatment with 1 µg/mL and 0.1 µg/mL cardanol compared to untreated cells. The percentages of wound closure after treatment with 10, 1, and 0.1 of cardanol were 42.36 ± 12.64%, 64.84 ± 7.11%, and 74.94 ± 5.92%, respectively, both after 24 h, compared with the untreated control (41.79 ± 3.47%). However, the percentages of wound closure after treatment with 10, 1, and 0.1 of cardanol were 79.3 ± 6.9%, 89.7 ± 5.3%, and 89.1 ± 1.9%, respectively, both after 48 h, compared with the untreated control (69.61 ± 5.64) (Figure 5). These findings highlight the wound-healing properties of cardanol on HaCaT cells, even at low doses.

### 3.5. Proliferative Effect of Cardanol on HaCaT Cells

The effect of cardanol on HaCaT cell proliferation was evaluated at 10, 1, and 0.1 µg/mL cardanol (Figure 7). Cardanol promoted cell proliferation compared to the control group at 1 and 0.1 µg/mL concentrations, highlighting the marked cell proliferation *p* < 0.05.

### 3.6. Modulatory Effect of Cardanol on the Cellular Fatty Acid Profile

To investigate the alterations of fatty acid profiles of HaCaT cells after supplementation with 1 µg/mL cardanol, the absolute amounts of fatty acids were determined using gas chromatography in comparison with the profiles of cells treated with the basic medium supplemented with 1 µg/mL castor oil (Table 1). Cardanol increased significantly the total amounts of ω-3 (*p* < 0.01), ω-6 (*p* < 0.001), ω-9 (*p* < 0.01), saturated fatty acids (*p* < 0.001), and fatty acids in general (*p* < 0.01). As examples, cardanol increased palmitic acid (*p* < 0.001), oleic acid (*p* < 0.01), linoleic acid (*p* < 0.001), arachidonic acid (*p* < 0.001), eicosapentaenoic acid (*p* < 0.01), docosahexaenoic acid (*p* < 0.01), and docosapenatenoic acid (*p* < 0.01). However, cardanol supplementation did not lead to a significant effect on the ω-6/ω-3 ratio under normal non-inflammatory conditions. While the ω-6/ω-3 ratio following cardanol treatment under inflammation is not provided here, the observed increase in anti-inflammatory fatty acids such as oleic acid, eicosapentaenoic acid, docosahexaenoic acid, and docosapentaenoic acid suggests a potential modulation of the ω-6/ω-3 balance under inflammation.

## 4. Discussion

The present study explores the structural analysis and biological activities of cardanol, a product from the technical refining of CNSL. CNSL is a mixture of phenolics with unsaturated C_15_-side chains, providing unique chemical properties, such as antimicrobial, antioxidant, and enzyme-inhibiting effects [25,26,27].

Here, we describe in detail by NMR spectroscopy the chemical composition of a technical cardanol sample as a mixture of 3-n-pentadecylphenols carrying one double bond at the 8′–9′ position, two double bonds at the 8′–9′ and 11′–12′ positions, or three double bonds at the 8′–9′, 11′–12′, and 14′–15′ positions, each in the cis configuration. In previous NMR studies, the monoene and diene isomers of anacardic acid were described by Suo et al. (2012), while the spectra for the triene could not be shown [28]. However, Morais and others described the three aliphatic side chains for anacardic acid [29]. Our detailed NMR analysis now affords the assignments of most ^1^H and ^13^C NMR signals for cardanol. To the best of our knowledge, this is the first comprehensive NMR analysis of cardanol. The relative fractions of the monoene, diene, and triene in the commercial product were determined by quantitative NMR spectroscopy as 42, 22, and 36% for the monoene, diene, and triene, respectively. However, the relative percentages of these components may depend on the underlying technical processes during CNSL extraction and on the plant species. This has to be taken into account when discussing the biological activities of cardanol, such as the antimicrobial effects against Gram-positive bacteria [30].

Generally, numerous biological activities of CNSL have been reported, including anti-inflammatory effects [31,32], antitumor properties [33], antioxidant activity [34,35,36], and multiple antimicrobial effects [37,38]. However, there is limited research on these substances’ safety and further potential therapeutic applications, particularly for wound healing. This gap in the literature prompted our investigation into the safety of cardanol on HaCaT cells, where we utilized both MTT and comet assays to assess its cytotoxicity and genotoxicity. The MTT assay demonstrated that cardanol is safe for HaCaT cells at 10 µg/mL or lower concentrations. A significant cytotoxic effect was observed only at higher doses, specifically at 100 µg/mL. This cytotoxicity may be attributed to the irritant properties of cardanol. This aligns with observations by Andonaba and others who documented skin damage and esthetic effects among workers at a cashew nut shelling [21]. The MTT assay is widely utilized to evaluate the cytotoxic effects of bioactive substances in vitro, as mitochondrial activity correlates directly with the number of viable cells. Given these insights, it is crucial to consider carefully the dosage of cardanol in clinical applications. Furthermore, to ensure translational relevance, we recommend conducting additional safety and efficacy studies using in vivo models and the intended target species to validate these observations.

Genotoxicity refers to harmful genetic changes, which include gene mutations, structural chromosomal aberrations, and recombination, all of which can be induced by genotoxic agents [39,40]. Notably, assessing genotoxicity is a prerequisite for registering new drugs [41]. Several methods are available for evaluating the genotoxicity of bioactive substances in vitro. These methods include the micronucleus assay, cytokinesis-block micronucleus cytometry assays, detection of 8-hydroxydeoxyguanosine DNA adducts, the hypoxanthine-guanine phosphoribosyltransferase forward mutation assay, and single-cell gel electrophoresis (also known as the comet assay) [42,43,44,45].

In this study, we have used the comet assay to assess the genotoxicity of cardanol in HaCaT cells. Our findings indicated that low doses of cardanol (1 and 0.1 µg/mL) did not affect DNA fragmentation significantly compared to the untreated control (Figure 4). However, significant genotoxicity was observed in HaCaT cells treated with a >10 µg/mL concentration. It is important to note that there are three known mechanisms by which genotoxins operate: carcinogens, mutagens, and teratogens [46]. Further studies are needed to investigate the specific mechanisms behind the genotoxic effects of cardanol with high doses.

Cardanol has a high phenolic content, which gives it notable antimicrobial and anti-inflammatory properties, making it a strong candidate for wound care. Its antimicrobial activity is particularly beneficial for maintaining a sterile environment around wounds, while its anti-inflammatory properties help reduce excessive inflammation that can hinder the healing process. After an injury, the body triggers a series of cellular responses to restore the damaged skin barrier [47]. The initial inflammatory phase involves recruiting immune cells to the wound site, releasing cytokines that help prevent infection and initiate healing [48]. The proliferative phase occurs, where cells such as keratinocytes and fibroblasts migrate to the wound site, proliferate, and begin forming new tissue [49].

In this study, we also investigated the effects of cardanol on wound-healing responses in HaCaT cells. We found that cardanol enhances the wound repair capabilities of HaCaT cells at concentrations of 1 µg/mL and 0.1 µg/mL compared to untreated cells. Both cell migration and proliferation play critical roles in wound healing, but their significance varies depending on the timing of the process. To differentiate between these two aspects within our designated timeframe of 24 h, we pre-treated the cells with mitomycin C (10 µg/mL) before inducing a scratch, which inhibited cell proliferation. As a result, we observed that wound closure under the influence of cardanol was significantly impacted. The activation of keratinocyte migration and proliferation upon skin damage is crucial for re-epithelialization, essential for effective wound healing [50].

The cellular composition of fatty acids results from a complex interplay of phenomena that ensure optimal cell viability under various conditions. Our findings demonstrate that the supplementation of HaCaT cells with cardanol at a 1 µg/mL concentration modulates the cellular fatty acid composition. Specifically, cardanol increased total polyunsaturated fatty acids (PUFAs, including ω-3, ω-6, and ω-9) and the total amount of saturated fatty acids. PUFAs, which contain two or more double bonds within their acyl chains, are linked to membrane domains’ structural and physical properties and are recognized for their health benefits [51]. The balance between ω-3 and ω-6 PUFAs is crucial for maintaining homeostasis in the body [52,53]. Excessive levels of ω-6 PUFAs, such as arachidonic acid (AA), can produce pro-inflammatory eicosanoids [54,55]. These substances contribute to thrombus and atheroma formation [56], allergic reactions [57], inflammatory disorders [58], excessive cell proliferation [59], and a hyperactive endocannabinoid system [60], which may increase appetite and food intake, potentially leading to weight gain and obesity. In contrast, ω-3 PUFAs, such as EPA and DHA, possess anti-inflammatory [61], anti-aggregatory [62], vasodilatory [63], and broncho dilatory properties [64]. These benefits help to resolve inflammation, modulate vascular markers, and reduce the risk of cancer and cardiovascular diseases [62]. A favorable ω-6/ω-3 ratio is associated with reduced inflammation and improved skin health [65]. For instance, ω-3 fatty acids like EPA and DHA have been linked to anti-inflammatory effects in skin cells, potentially reducing the risk of inflammatory skin diseases [66]. Our findings indicate that cardanol could influence the structure and function of the cell membranes, thereby enhancing cellular responses to injury and inflammation. Additional studies on the modulatory effect of cardanol on fatty acids and the ω-6/ω-3 ratio under inflammatory conditions are still needed.

## 5. Conclusions

The composition of a technical cardanol sample was determined in detail by NMR spectroscopy as a mixture of 3-n-pentadecylphenols carrying an 8′-monoene (42%), an 8′,11′-diene (22%), or an 8′,11′,14′-triene (36%) side chain. For potential pharmaceutical usages, this cardanol appeared safe at doses ≤ 1 µg/mL, as shown by MTT and comet assays with HaCaT cells. Surprisingly, cardanol showed wound-healing properties in the scratch assay using HaCaT cells and promoted their proliferation. Together, these findings underscore the high potential of cardanol for skin health applications, thereby enhancing the vital cellular processes involved in tissue repair. It should be noted that all experiments in this study were conducted using a single batch of commercial cardanol. Batch-to-batch variability might arise due to potential differences in purity, composition, or storage conditions. Further studies are required to explore its broader therapeutic usages and its formulations for clinical applications under varied conditions. It is also essential to assess its safety more rigorously with other cell types and animal models. This could finally establish cardanol, a bulk product in the technical processing of cashew nuts, as a beneficial product for medicinal and lifestyle applications.

## Figures and Tables

**Figure 1 metabolites-15-00083-f001:**
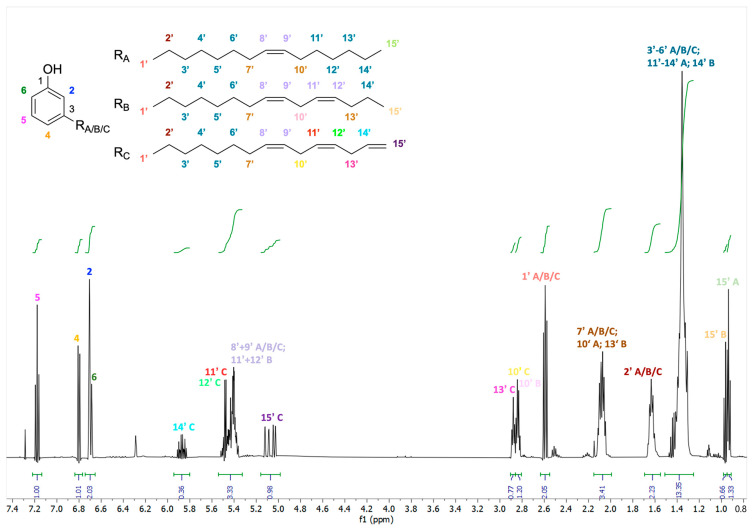
Chemical structures of the main components A, B, and C in cardanol and the ^1^H NMR spectrum of cardanol at 500 MHz in CDCl_3_. Before Fourier transformation, the FID was multiplied with a mild Gaussian function.

**Figure 2 metabolites-15-00083-f002:**
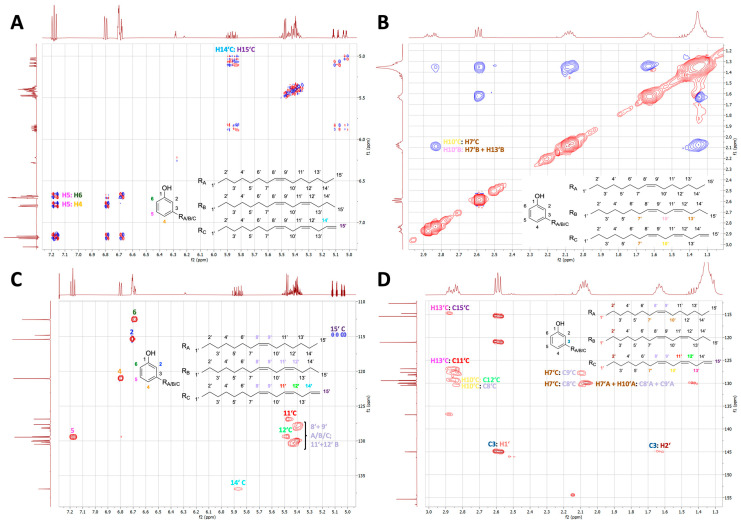
Relevant sections of two-dimensional double-quantum-filtered COSY (**A**), NOESY (**B**), DEPT-edited HSQC (**C**), and HMBC spectra (**D**). In the phase-sensitive COSY, NOESY, and HSQC experiments, the amplitudes of the signals are indicated by red and blue. In the NOESY spectrum, the negative NOE peaks are displayed in blue colors. In the HSQC spectrum, the phases were adjusted as positive signals (red colored) for CH and CH_3_ atoms and as negative signals (blue colored) for CH_2_ atoms.

**Figure 3 metabolites-15-00083-f003:**
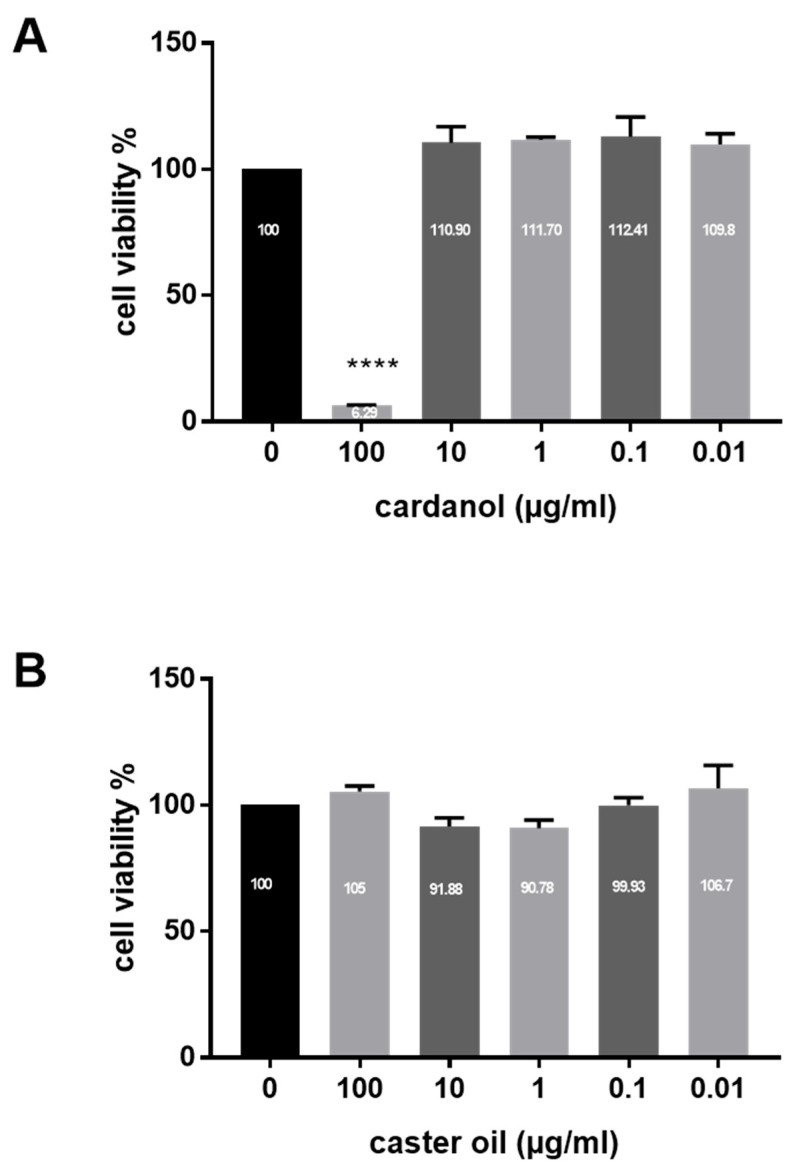
Cytotoxicity of cardanol on HaCaT keratinocytes. (**A**) With cardanol, (**B**) with basic medium supplemented with the emulsifier, and castor oil. **** *p* < 0.0001. The values represent mean ± SE (*n* = 3).

**Figure 4 metabolites-15-00083-f004:**
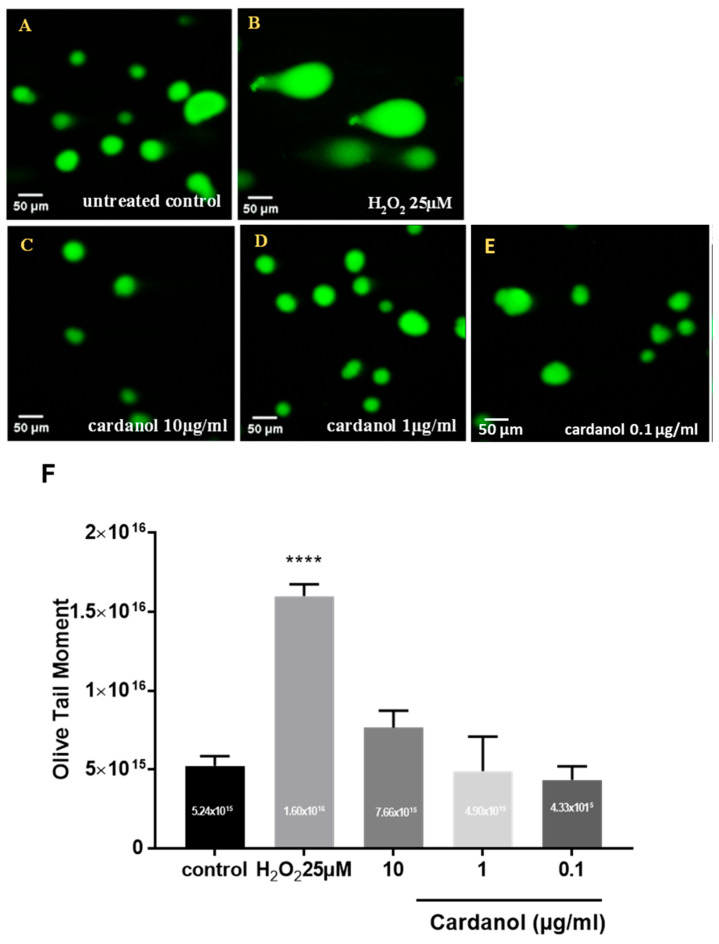
Comet assay for evaluation of the toxicity of the cardanol on HaCaT keratinocytes. DNA damage is visualized in the form of comets in keratinocytes. (**A**) Untreated control, (**B**) cells treated with 25 µM H_2_O_2_ for 30 min, (**C**–**E**) cells treated with 10, 1, and 0.1 of cardanol (magnification ×40), (**F**) DNA damage based on the automated olive tail moment calculation in HaCaT cells using the OpenComet tool in Image J Software. The values represent mean ± SE (*n* = 3) (**** *p* < 0.0001).

**Figure 5 metabolites-15-00083-f005:**
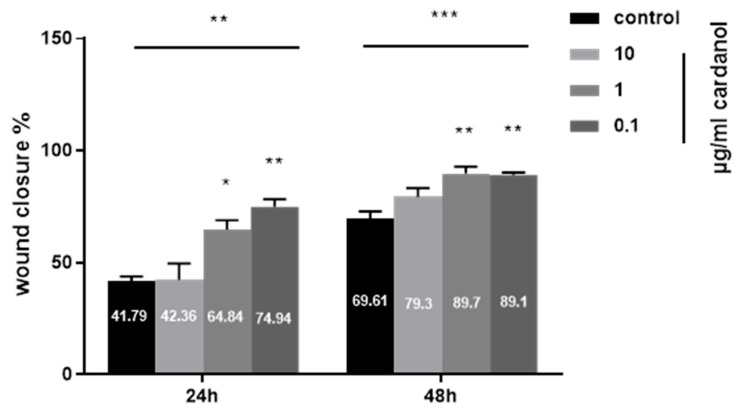
Effect of cardanol on the wound healing of HaCaT cells. Data from three independent experiments. Standard deviation bars denote SE. (*)—statistically significant differences compared to the control (* *p* < 0.01, ** *p* < 0.001, and *** *p* < 0.0001).

**Figure 6 metabolites-15-00083-f006:**
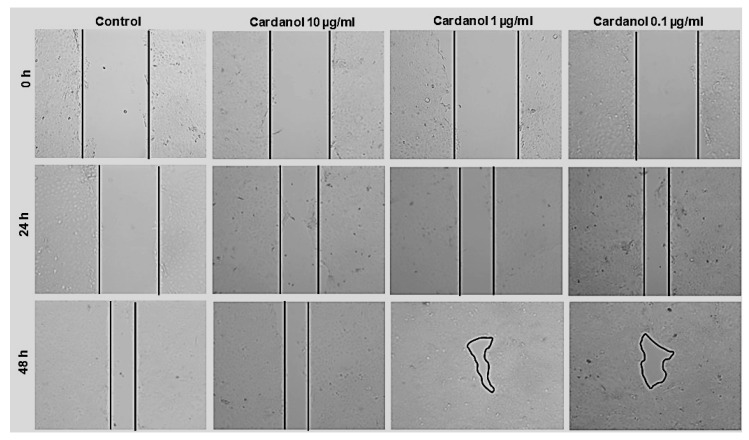
Effect of cardanol on the wound healing of HaCaT cells. HaCaT cells were treated with mitomycin c and various concentrations of cardanol for 0 h, 24 h, and 48 h. Cell migration was visualized via phase-contrast microscopy.

**Figure 7 metabolites-15-00083-f007:**
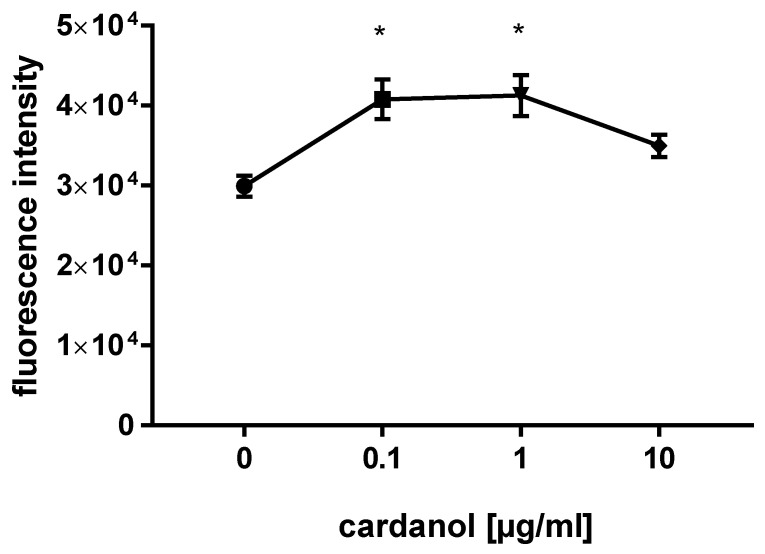
Effect of cardanol on the proliferation of HaCaT cells at a range of concentrations (0.1, 1, 10 µg/mL) over 48 h (*n* = 3); data are presented as means  ±  SE; * *p* < 0.05 (all one-way analysis of variance).

**Table 1 metabolites-15-00083-t001:** Modulatory effect of cardanol on the cellular fatty acid profile of HaCaT cells using gas chromatography.

Fatty Acid	HaCaT Grown in Basic Medium ^1^	HaCaT Grown in Medium with Cardanol ^2^	*p*-Value
Palmitic acid	194.00 ± 6.37	249.75 ± 8.43 **	*p* < 0.001
Oleic acid	323.35 ± 11.06	420.44 ± 23.44 *	*p* < 0.01
Linoleic acid	19.82 ± 0.24	25.92 ± 1.17 **	*p* < 0.001
Arachidonic acid	46.98 ± 0.14	60.04 ± 2.15 **	*p* < 0.001
Eicosapentaenoic acid	3.12 ± 0.29	4.10 ± 0.70	*p* < 0.01
Docosahexaenoic acid	23.53 ± 0.74	32.33 ± 1.92 *	*p* < 0.01
Docosapenatenoic acid	23.22 ± 0.67	30.24 ± 1.93 *	*p* < 0.01
Total ω6 fatty acids	89.57 ± 0.75	114.42 ± 5.13 **	*p* < 0.001
Total ω3 fatty acids	51.65 ± 1.52	68.56 ± 4.49 *	*p* < 0.01
Total ω9 fatty acids	357.25 ± 12.40	463.23 ± 72.28 *	*p* < 0.01
Total saturated fatty acids	295.19 ± 11.07	382.22 ± 14.35 **	*p* < 0.001

^1^ Basic medium = DMEM containing 1 µg/mL castor oil, ^2^ basic medium containing cardanol 40% in castor oil, supplemented at a concentration of 1 µg/mL. Asterisks (*,**) indicate significant differences compared with the basic medium.

## Data Availability

The data that support the findings of this study are available from the corresponding author upon reasonable request.

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
