# Peer review of "Structural Analysis of Cardanol and Its Biological Activities on Human Keratinocyte Cells"

_metabolites, 2025, doi:10.3390/metabo15020083_

Round 1
Reviewer 1 Report
Comments and Suggestions for Authors
The reviewed article is devoted to really relevant issue, - analyzing the composition of a technical cardanol sample from cashew nutshell liquid (CNSL), researching its safety profile and therapeutic potential for human skin cells.
I believe that the authors have performed a considerable amount of research, using advanced approaches, and provided statistical data processing with software tools to confirm the validity of the results obtained.
The presented article is designed according to the stylistic standards for a scientific text and in accordance with the publisher requirements; I am recommended this manuscript for publication with minor modifications:
In the "Introduction" section, in lines 33-37, overly general phrases are used that are not specifically relevant to this study.
In the “Materials and Methods” section, explain how the cardanol sample was obtained.
In subsection 3.6, explain how cardanol affected the w-6 to w-3 fatty acids ratio, and how it correlates with the recommended value for the lipid layer of skin cells.
Author Response
The reviewed article is devoted to really relevant issue, - analyzing the composition of a technical cardanol sample from cashew nutshell liquid (CNSL), researching its safety profile and therapeutic potential for human skin cells. I believe that the authors have performed a considerable amount of research, using advanced approaches, and provided statistical data processing with software tools to confirm the validity of the results obtained. The presented article is designed according to the stylistic standards for a scientific text and in accordance with the publisher requirements;
The authors appreciate your comments, which have greatly improved our manuscript.
I am recommended this manuscript for publication with minor modifications:
- In the "Introduction" section, in lines 33-37, overly general phrases are used that are not specifically relevant to this study.
These phrases are deleted. Please see line 32.
- In the “Materials and Methods” section, explain how the cardanol sample was obtained.
Available information about the samples has been added; please see Materials and Methods section 2.1, lines 92-94.
According to the manufacturer, cardanol is obtained by vacuum distillation of CNSL. In this study, we tested a sample obtained from only one batch. Therefore, we have included a statement in the Conclusion section acknowledging the possibility of such variations and their potential impact on the reproducibility of the results if different batches used. “It should be noted that all experiments in this study were conducted using a single batch of commercial cardanol. Batch-to-batch variability might arise due to potential differences in purity, composition, or storage conditions.”. Please see the Conclusion, lines 428-430.
- In subsection 3.6, explain how Cardanol affected the w-6 to w-3 fatty acids ratio, and how it correlates with the recommended value for the lipid layer of skin cells.
Thank you very much for this comment. Indeed, cardanol supplementation leads to a non-significant effect on the ω-6/ω-3 ratio under normal non-inflammatory conditions.
We have added in the Result section 3.6, lines 312-317 the following phrases: “However, cardanol supplementation did not lead to a significant effect on the ω-6/ω-3 ratio under normal non-inflammatory conditions. While the ω-6/ω-3 ratio following cardanol treatment under inflammation is not provided here, the observed increase in anti-inflammatory fatty acids such as oleic acid, eicosapentaenoic acid, docosahexaenoic acid, and docosapentaenoic acid suggests a potential modulation of the ω-6/ω-3 balance under inflammation.”
Also, in Discussion lines 412-418, we added the following phrases. “A favorable ω-6/ω-3 ratio is associated with reduced inflammation and improved skin health [65]. For instance, ω-3 fatty acids like EPA and DHA have been linked to anti-inflammatory effects in skin cells, potentially reducing the risk of inflammatory skin diseases [66]. Our findings indicate that cardanol could influence the structure and function of the cell membranes, thereby enhancing cellular responses to injury and inflammation. Additional studies on the modulatory effect of cardanol on fatty acids and the ω-6/ω-3 ratio under inflammatory conditions are still needed.”.
Reviewer 2 Report
Comments and Suggestions for Authors
In the current manuscript, the authors characterized a technical cardanol sample using NMR, and profiled its chemical composition. They then extensively tested its influence on a keratinocyte cell line for toxicity, growth characteristic and lipid profiles. The experiments were well designed and executed, and the paper was well written. I have a few suggestions:
1. My biggest concern is the way that the data from the cell experiments were presented and their statistical significance, and as a consequence, the lack of a strong arguments built upon many experiments that were done. For instance, lines 288-293: the variations within the datasets were overly large, making it difficult to be conclusive comparing to the controls. Also, for example, the data from the proliferation effects experiment (section 3.5) and fatty acid profiling (section 3.6) were quite noisy. The author should consider re-analyzing the cell data, and perhaps trying to incorporate more data points into the analysis. For bar charts, please plot the individual values within the bars.
2. For the cell viability experiment, the authors should establish a more granular kill curve for the range between 10 and 100.
3. HaCaT represents an immortalized format of keratinocytes, for assessment of toxicity tolerance and morphology characterization, an orthogonal cell type (e.g. K38 or differentiated HaCaT) should be tested alongside to be conclusive on the claim of cardanol’s potential therapeutic use. In light of the manuscript's current shape: the authors should also make modifications to the discussions in the Abstract and Discussions to qualify the experimental implications from this work to the cell line used (not keratinocyte as a tissue type).
4. The numbering of the figures were wrong after Fig 3 on page 7; another “Fig 3” popped up on page 10, it should be Fig 4, and the ensuing figures’ numbers should be adjusted accordingly.
5. Line 288, within Fig 4: remove the redundant “F”.
6. The authors should acknowledge the potential possibility of variations in the experimental outcomes caused by batch differences of the cardanol.
7. It may be helpful to consolidate Figures 1-3 into 1 or 2 figures. And it may be helpful to move Table 1 into the supplementary dataset.
8. Lines 294-304, 315-319, 324-334: please add a brief summary to each section.
Author Response
- My biggest concern is the way that the data from the cell experiments were presented and their statistical significance, and as a consequence, the lack of a strong arguments built upon many experiments that were done. For instance, lines 288-293: the variations within the datasets were overly large, making it difficult to be conclusive comparing to the controls. Also, for example, the data from the proliferation effects experiment (section 3.5) and fatty acid profiling (section 3.6) were quite noisy. The author should consider re-analyzing the cell data, and perhaps trying to incorporate more data points into the analysis. For bar charts, please plot the individual values within the bars.
The authors appreciate the reviewer´s comments that improved our manuscript. The graphs have been revised. We realized that the SD was added by default. We revised the figures with the SE instead of SD to clarify the statistical difference. We strongly agree with this comment that the SD is quite high. However, Comet assay is a sensitive technique for measuring DNA damage. To address the concern about variability, we used one-way ANOVA with the Tukey test as a post-analysis test to compare the control with the H2O2-stimulated cells, as well as the cardanol-supplemented cells. The experiment was repeated three times in double replicates. Individual values were plotted within the bar charts as kindly recommended.
Figure 7 has been revised also. The order of the concentrations should be 0.1, 1 and 100 from left to right (not 100, 1 and 0.1). The proliferation assay results clearly demonstrate the proliferative effect of very low doses of cardanol, aligning with the primary objective of this experiment. The statistic was revised using one-way ANOVA analysis. Cardanol at 0.1, 1 µg/mL induced a significant proliferation effect of HaCaT cells.
- For the cell viability experiment, the authors should establish a more granular kill curve for the range between 10 and 100.
We strongly agree with the comment that a more granular curve will provide additional precision, but we focused on the ranges most relevant to the study's objectives. Since cardanol at <=10 µg (0.1, 1, and 10 µg/mL) is safe, we focused on these concentrations in performing the biological studies, including comet assay, scratch assay, and proliferation test. We think that data with more than 10 µg/mL would not benefit our study. However, future studies will consider a more detailed dose-response analysis in this range.
- HaCaT represents an immortalized format of keratinocytes, for assessment of toxicity tolerance and morphology characterization, an orthogonal cell type (e.g. K38 or differentiated HaCaT) should be tested alongside to be conclusive on the claim of Cardanol’s potential therapeutic use. In light of the manuscript's current shape: the authors should also make modifications to the discussions in the Abstract and Discussions to qualify the experimental implications from this work to the cell line used (not keratinocyte as a tissue type).
We fully agree. We better referred the text to the usage of HaCaT cells with experimental implications and limitations from this work. For example, in the Abstract, we added: “Further studies are necessary to explore additional potential applications of cardanol, to refine its formulations for clinical use, and to ensure its safety and action in other target cells and species.”
- The numbering of the figures were wrong after Fig 3 on page 7; another “Fig 3” popped up on page 10, it should be Fig 4, and the ensuing figures’ numbers should be adjusted accordinglyL
The numbering of figures and tables was corrected.
- Line 288, within Fig 4: remove the redundant “F”.
It was removed as kindly recommended.
- The authors should acknowledge the potential possibility of variations in the experimental outcomes caused by batch differences of the Cardanol.
We have included a statement in the Discussion section lines 428-432 acknowledging the possibility of such variations and their potential impact on the reproducibility of the results. “It should be noted that all experiments in this study were conducted using a single batch of commercial cardanol. Batch-to-batch variability might arise due to potential differences in purity, composition, or storage conditions. Further studies are required to explore its broader therapeutic usages and its formulations for clinical applications under varied conditions.”
- It may be helpful to consolidate Figures 1-3 into 1 or 2 figures. And it may be helpful to move Table 1 into the supplementary dataset.
The original Figures 1 and 2 were merged into Figure 1, as kindly recommended. Also, Table 1 was moved into the supplementary dataset.
- Lines 294-304, 315-319, 324-334: please add a brief summary to each section.
It was done as kindly recommended; please see sections 3.4 and 3.6.
Reviewer 3 Report
Comments and Suggestions for Authors
In this article, Basiouni et.al A commercial sample of cardanol was analyzed using nuclear magnetic resonance (NMR) spectroscopy, which revealed a mixture with different product ratios. The cytotoxic effect on the human keratinocyte cell line (HaCaT cells) was only observed at high concentrations of cardanol (> 10 micrograms/ml), which led to a significant decrease in cell viability. The article may be of general interest, and I suggest publishing it after minor revision. In particular, the authors have put a lot of effort into studying the reaction and obtaining stereochemistry products of all the compounds obtained. In Figure 1, highlight what each signal corresponds to, preferably in color. Figure 3. The experimental error is not quite visible. What part did it make up? In my opinion, one very important point remains unclear: are the products obtained the only ones and how can their formation be explained? Have the conditions of receipt been optimized?
Author Response
In this article, Basiouni et.al A commercial sample of Cardanol was analyzed using nuclear magnetic resonance (NMR) spectroscopy, which revealed a mixture with different product ratios. The cytotoxic effect on the human keratinocyte cellline (HaCaTcells) was only observed at high concentrations of Cardanol(>10micrograms/ml),which led to a significant decrease in cell viability. The article maybe of general interest, and I suggest publishing it after minor revision. In particular, the authors have put a lot of effort into studying the reaction and obtaining stereochemistry products of all the compounds obtained.
The authors appreciate your comments, which have greatly improved our manuscript.
- In Figure 1, highlight what each signal corresponds to, preferably in color-
Figure 1 was revised as kindly recommended.
- Figure The experimental error is not quite visible. What part did it make up?
Thanks a lot for this comment, the MTT assay used in this study is a well-established and normalized to the control, resulting in a non-visible experimental error.
- In my opinion, one very important point remains unclear: are the products obtained the only ones, and how can their formation be explained? Have the conditions of receipt been optimized? Have the conditions of receipt been optimized?
We appreciate this comment. Cardanol (commercial product) was obtained from Sona Meta Chem Pvt. Ltd., Pondicherry, India. According to the manufacturer, this cardanol is obtained by vacuum distillation of CNSL. In our study, we tested a sample obtained from only one batch. Therefore, we have included a statement in the Discussion section, lines 428-432, acknowledging the possibility of such variations and their potential impact on the reproducibility of the results. “It should be noted that all experiments in this study were conducted using a single batch of commercial cardanol. Batch-to-batch variability might arise due to potential differences in purity, composition, or storage conditions. Further studies are required to explore its broader therapeutic usages and its formulations for clinical applications under varied conditions.”